# Position: Agentic AI Orchestration Should Be Bayes-Consistent

Theodore Papamarkou [1][2]  Pierre Alquier [3]  Matthias Bauer  Wray Buntine [4]  Andrew Davison [5]
Gintare Karolina Dziugaite [6]  Maurizio Filippone [7]  Andrew Y. K. Foong [8]  Vincent Fortuin [9][10]
Dimitris Fouskakis [2]  Jes Frellsen [11]  Eyke Hüllermeier [12]  Theofanis Karaletsos [13]  Mohammad Emtiyaz Khan [14]
Nikita Kotelevskii [15]  Salem Lahlou [15]  Yingzhen Li [5]  Fang Liu [16]  Clare Lyle  Thomas Möllenhoff [14]
Konstantina Palla [17]  Maxim Panov [15]  Yusuf Sale [12]  Kajetan Schweighofer [18]  Artem Shelmanov [15]
Siddharth Swaroop [19]  Martin Trapp [20]  Willem Waegeman [21]  Andrew Gordon Wilson [22]  Alexey Zaytsev [23]

## Abstract

LLMs excel at predictive tasks and complex reasoning tasks, but many high-value deployments rely on decisions under uncertainty, for example, which tool to call, which expert to consult, or how many resources to invest. While the usefulness and feasibility of Bayesian approaches remain unclear for LLM inference, this position paper argues that the control layer of an agentic AI system (that orchestrates LLMs and tools) is a clear case where Bayesian principles should shine. Bayesian decision theory provides a framework for agentic systems that can help to maintain beliefs over task-relevant latent quantities, to update these beliefs from observed agentic and human-AI interactions, and to choose actions. Making LLMs themselves explicitly Bayesian belief-updating engines remains computationally intensive and conceptually nontrivial as a general modeling target. In contrast, this paper argues that coherent decision-making requires Bayesian principles at the orchestration level of the agentic system, not necessarily the LLM agent parameters. This paper articulates practical properties for Bayesian control that fit modern agentic AI systems and human-AI collaboration, and provides concrete examples and de-

sign patterns to illustrate how calibrated beliefs and utility-aware policies can improve agentic AI orchestration.

## 1. Introduction

Large language models (LLMs) have become a powerful tool for building modern AI applications (Brown et al., 2020). They use high-dimensional distributions over text sequences and multimodal inputs, and are capable of solving a wide range of tasks that were previously believed to require human-level intelligence, simply by conditioning on input prompts. However, in many practical deployments, especially in high-stakes and safety-critical settings, a key bottleneck for LLM-based systems is not producing a plausible response, but *making decisions under uncertainty based on the available information* (Liu et al., 2025). Such decisions include stopping, asking clarification questions, and tool-selection via routing, as well as budget allocation that trades accuracy against cost, latency, and risk (Schick et al., 2023; Ong et al., 2025; Suri et al., 2025; Wen et al., 2025). This shift from sequence prediction to *agentic behavior* (Yao et al., 2023) changes the evaluation target.

Bayesian decision theory (Berger, 1985; Smith, 2010) provides a principled framework for agentic settings. It formalizes decision-making in terms of maintaining beliefs over task-relevant latent variables, updating those beliefs as new evidence arrives, and selecting actions that maximize posterior expected utility or, equivalently, acquiring additional information only when its expected value exceeds its cost. These elements align with the control problems faced by agentic AI systems, where routing, stopping, escalation, and budget allocation decisions must account for uncertainty and asymmetric costs.

This framing suggests two broad routes to Bayes-optimal behavior in LLM-based systems. One route is to make LLMs themselves Bayesian belief-updating engines that maintain and update full posteriors over model parameters. Another

---

[1]PolyShape, Greece. [2]NTUA, Greece. [3]ESSEC Business School, Singapore. [4]VinUniversity, Vietnam. [5]Imperial College London, UK. [6]Mila - Quebec AI Institute, Canada. [7]KAUST, Saudi Arabia. [8]Mayo Clinic, USA. [9]TU Nuremberg, Germany. [10]Helmholtz AI, Germany. [11]Technical University of Denmark, Denmark. [12]MCML, LMU Munich, Germany. [13]Pyramidal Inc., USA. [14]RIKEN, Japan. [15]MBZUAI, UAE. [16]University of Notre Dame, USA. [17]Spotify, UK. [18]Cognizant AI Lab, USA. [19]University College London, UK. [20]KTH Royal Institute of Technology, Sweden. [21]Ghent University, Belgium. [22]New York University, USA. [23]BIMSA, China. Correspondence to: Theodore Papamarkou <theodore@polyshape.com>.

*Proceedings of the $43^{rd}$ International Conference on Machine Learning*, Seoul, South Korea. PMLR 306, 2026. Copyright 2026 by the author(s).

route is to treat LLMs as powerful but non-Bayesian components and to embed Bayesian structure into the *control and orchestration layer* that determines how models and tools are queried and composed.

The first route remains out of reach in realistic systems. Even when LLMs exhibit Bayesian-like behavior in restricted regimes, they may not always reliably satisfy the properties of full Bayesian belief-updating agents (Falck et al., 2024). It has also been argued that posteriors over parameters can be a poor mechanism for representing epistemic uncertainty in highly overparameterized models (Kirsch, 2025).

Due to these issues, the usefulness of Bayesian principles remain unclear for LLM inference. These considerations motivate us to think of cases that are, in our opinion, more realistic, feasible, and decision-relevant. Such cases may include maintaining a Bayesian model over the latent variables of the underlying agentic AI system to encode hypotheses, tool uncertainty, and agent reliability.

> **Position.** The role of Bayesian reasoning for LLMs and agentic AI becomes clear once prediction is separated from decision making. An LLM is a predictive model that represents high-dimensional distributions over sequences and can estimate predictive probabilities for task-level events. An agentic AI system, in contrast, selects and executes actions in an environment, with LLM-based predictive components serving as input sources within the system's decision process. In agentic AI systems, Bayesian reasoning is both feasible and valuable for reliable control and composition. **This position paper therefore argues that LLMs need not be explicitly Bayesian, while AI systems that orchestrate LLM agents should be making decisions at least consistent with Bayesian reasoning**.

This paper proposes to design agentic AI systems in which LLMs and tools remain black-box predictors. A Bayesian controller layer maintains beliefs over task-relevant latent quantities and implements expected-utility (DeGroot, 2004; Smith, 2010) or value-of-information policies (Stratonovich, 1965; Howard, 1966; Belavkin, 2009; Gelman et al., 2013) for selecting actions, such as querying, orchestrating, routing, or stopping. The controller tracks and, after observing evidence, updates posterior distributions over a set of latent task variables. It triggers another tool call only when the expected improvement in outcome outweighs its cost.

## 2. Rationale For The Position

Bayesian reasoning can, in principle, be placed at many levels of an AI system, namely, inside model training, in the design of inference procedures, or in the control logic that regulates the use of models and tools. In this position paper, the focus is on the control level, with Bayesian structure treated as a property of agentic systems that select, use, and compose LLM-based components, rather than as an internal requirement on the LLMs themselves.

Here, a *Bayesian agentic system* is defined by its control layer. It maintains a belief state over task-relevant latent variables and selects actions by maximizing posterior expected utility or by comparing the value of information against cost. This differs from one possible notion of a *Bayesian LLM*, which implements Bayesian belief updating internally and produces token probabilities by integrating the next-token distribution (conditional on the preceding context and the LLM parameters) over the posterior distribution on the LLM's parameters. This approach to Bayesian LLMs has been prohibitively expensive without major breakthroughs, and we propose to formulate Bayesian approaches in the belief state and control policy of the agentic system.

A central motivation for placing Bayesian structure in the control layer, rather than inside the LLM, is that the type of uncertainty expressed by LLMs does not coincide with the epistemic uncertainty that drives agentic decisions, and the internal updating behavior of LLMs is not, in general, Bayesian. Token-level predictive distributions can be sharp while the system remains uncertain about the task-level latent variable that matters for action selection, and conversely, a diffuse next-token distribution can coexist with high confidence about the underlying semantic answer or subsequent outcome. This discrepancy complicates any attempt to use token-level probabilities as the belief state for control (Abbasi-Yadkori et al., 2024; Bakman et al., 2025), first recognized as the issue of syntactic versus semantic uncertainty (Kuhn et al., 2023; Aichberger et al., 2025).

Moreover, empirical diagnostics implied by posterior predictive processes under standard assumptions, such as exchangeability and martingale constraints, are often violated by in-context predictions of pretrained LLMs (Falck et al., 2024), with partial mitigation depending on inference procedures and prompting (Jayasekera et al., 2025). Related evidence suggests that even when Bayesian behavior holds only in expectation or under special regimes, individual LLM outputs can depart systematically from Bayesian properties (Chlon et al., 2025; Atwell et al., 2026), and approximate Bayesian neural networks can also fail sequential updating properties (Pituk et al., 2025). These considerations point toward a belief state over lower-dimensional, decision-relevant latent variables, updated through observation models calibrated against measurable outcomes, as a more defensible interface between uncertainty and action in agentic AI systems. For further context, Appendix A provides extended background and related work.

Decision making in general does not require an explicit prob-

abilistic belief state. Many reinforcement learning methods, for example, optimize expected return through value functions or policy gradients without maintaining a distribution over hypotheses or latent task variables. The claim behind Bayesian control is stronger. It asserts that when an agentic system acts under epistemic uncertainty and when actions have asymmetric or context-dependent costs, then a feasible approach is to represent uncertainty over a decision-relevant latent state. This approach requires updating uncertainty as observations arrive. Such a framework makes the decision criterion explicit and, therefore, forces uncertainty in the orchestration of the agentic AI system to be accounted for adaptively and robustly. Changes in the user, the task, or the tool ecosystem enter as evidence that shifts the posterior over the latent state. This shift consequently alters the optimal action of the agentic system.

If future agentic systems incorporate world models (Ha & Schmidhuber, 2018; LeCun, 2022), then Bayesian belief updating may become more naturally embedded within the model itself. The present position concerns today's LLM-centric agentic systems, where the Bayesian structure is most actionable at the agentic control layer.

It is useful to identify what this position implies for the design of practical agentic systems. To this end, the following outline presents seven desirable properties that can make Bayesian control compatible with modern software stacks, agentic AI systems, and human-AI collaboration.

1. *Reasoning about utilities and costs*. Utilities and costs, such as a user-privacy risk penalty and a tool-call cost, should be treated as modeling components, rather than as constants. A Bayesian control layer should represent utilities and costs as parameters or latent variables, place priors over them when appropriate, update them from feedback, and select actions by maximizing posterior expected utility while integrating over uncertainty in both task state and utility specification.

2. *Improved decision making with low overhead*. The Bayesian control layer should improve decision quality under cost constraints, with a fixed budget, while quantifying uncertainty. It should do so with low latency and low memory overhead relative to model inference, and it should yield fewer redundant tool calls and fewer incorrect or unsafe actions at a given target risk level.

3. *Efficient interaction history integration*. The framework should incorporate interaction history through Bayesian distillation, maintaining a belief state that serves as an approximately sufficient statistic of past exchanges rather than a full interaction history.

This yields bounded memory and computational cost while preserving informational relevance.

4. *Human-AI and multi-agent integration*. The framework should extend to systems where humans and AI agents interact, treating human feedback and inter-agent communication as probabilistic observations within the same Bayesian structure, thus enabling collective decision making.

5. *Industry alignment*. The Bayesian interface should build on typed agent schemas that echo the design philosophy of contemporary programming ecosystems used for agentic AI (e.g., TypeScript and Python), enabling ease of integration.

6. *Multimodal readiness*. Any agent that can provide probabilistic beliefs about task-level events should fit the framework, regardless of whether it operates on text, images, audio, or video.

7. *Accessibility without Bayesian expertise*. Users can interact only through simple controls, such as a confidence threshold or a cost scale, while all Bayesian updates occur internally and are not exposed in the interface.

A key limitation is that observation models derived from high-dimensional agent messages can be misspecified, and repeated tool calls can yield correlated evidence. The practical mitigation is to make updates conservative via recalibration from measurable outcomes, likelihood tempering, dependence-aware evidence pooling, and abstention or escalation when posterior confidence is fragile. Appendix B elaborates on these and other limitations and mitigations.

## 3. Alternative Views

**Bayesian inference for LLMs.** The primary goal of Bayesian deep learning (BDL) has been to train large neural networks using Bayesian methods (Papamarkou et al., 2024). Methods such as Laplace's method, mean-field inference, and sampling methods were all proposed for neural networks early on in the 90s (Peterson & Anderson, 1987; Mackay, 1992; Hinton & Van Camp, 1993; Neal, 1996; Saul et al., 1996). Following this endeavor, early BDL attempts focused on applying such Bayesian methods on top of existing training procedures (Graves, 2011; Blundell et al., 2015; Gal & Ghahramani, 2016; Mandt et al., 2017; Khan et al., 2018; Zhang et al., 2018; Maddox et al., 2019; Osawa et al., 2019; Shen et al., 2024). Another effort has been in using Bayesian principles to unify existing learning algorithms and generalization theories by drawing connections between PAC-Bayes, optimization, and learning theory (Dziugaite & Roy, 2017; Alquier & Ridgway, 2020; Hennig et al., 2022;

Lotfi et al., 2022; Khan & Rue, 2023; Lotfi et al., 2024).

**Bayes has not yet reshaped LLM training.** While all such efforts have had some impact, they have not yet seen major breakthroughs in improving the state-of-the-art for LLM training, for example, such as those recently obtained by second-order optimization (Anil et al., 2021; Jordan et al., 2024). If successful, these efforts can help identify the sources of uncertainty in LLMs and fix them by allocating additional compute and data. They could initiate a shift towards continual learning and adaptation based on Bayesian principles (Khan, 2025), and could help address sustainability concerns due to the heavy reliance on data, compute, and other training infrastructure. However, it remains unclear whether this is achievable in the near future and if it will lead to a new paradigm for LLM training.

**Bayesian LLM compatibility with our position.** Our position in this paper does not rely on the success or failure of such approaches. Even if LLMs contain some Bayesian elements, agentic control can still face an irreducibly decision-theoretic problem. The uncertainties may arise at the task level and can be utility-dependent. Accordingly, even if a fully Bayesian LLM induced uncertainty over task variables, decision-space modeling would remain a useful control object because it is aligned with the latent variables, costs, and actions on which orchestration is evaluated. Our position is that agentic AI systems should include Bayesian elements at such levels too. One worry could be that, without any Bayesian elements in the constituent LLMs, the agentic system can be miscalibrated. This could also hamper the Bayesian effort at the agentic control level. Our position in this paper does not reject such difficulties (see Appendix B). We believe that, despite such difficulties, this control-layer approach should be explored, and Bayesian principles can still be useful to some extent.

**Heuristic or prompting-based approaches.** Another alternative view is that there is no need for Bayesian modeling at the level of agents or their orchestration. Contemporary LLMs have general-purpose capabilities to code, translate, and reason. From this perspective, it is reasonable to expect that agentic systems can also be orchestrated through prompting strategies and workflows based on chain-of-thought (Wei et al., 2022), as in common current practice (Schick et al., 2023; Yao et al., 2023). Such approaches may well be implicitly Bayesian, but there is limited empirical evidence that their performance is fundamentally constrained by the absence of probabilistic structure, or that an explicit Bayesian orchestration layer is required to close any performance gap. This observation is especially plausible in short-horizon or low-stakes settings, where simple prompting heuristics perform well, and failures are easily corrected. The effectiveness of these approaches cannot be denied. We view these approaches primarily as practical engineering baselines and current defaults, rather than as an opposed stance. Our position is instead that the need for principled agentic orchestration becomes important as horizons lengthen, stakes increase, and tool ecosystems grow, because uncertainty accumulates, evidence sources become correlated, and cost trade-offs become more pronounced. Agentic orchestration is a non-trivial problem, involving decisions about problem decomposition, assignment of sub-tasks to agents, coordination of communication, and the management of uncertainty and partial information across agents. In these scenarios, decisions about routing and resource allocation are difficult to design when encoded only as fixed or handcrafted workflows, whereas a Bayesian control layer provides a belief state that supports adaptive outcome-based orchestration.

**Alternative decision frameworks.** Another alternative view questions whether Bayesian control is the right decision-theoretic formalism for agentic orchestration. From this perspective, the relevant problem is sequential decision making under uncertainty, and there exist approaches that do not require maintaining an explicit posterior belief state, including robust control, reinforcement-learning planning, bandit formulations based on confidence bounds, and objective-based orchestration (Shinn et al., 2023; Yao et al., 2023). In this view, uncertainty can be handled implicitly through exploration bonuses, worst-case guarantees, or repeated trial-and-error interaction, and the additional modeling burden of specifying latent variables, likelihoods, and priors may be seen as unnecessary. Our position is that Bayesian decision theory is advantageous when an orchestrator must trade off information gathering against costs and risks, when it must adaptively discover how to decompose a problem into sub-tasks, and when it must support abstention, escalation to a human expert, or tool selection based on a notion of expected value of information rather than on a fixed heuristic. In such settings, a belief state over task-level latent variables serves as an interface between evidence and action, and it allows the agentic system to couple uncertainty to utilities in a way that is difficult to express through purely reward-driven or worst-case procedures.

## 4. Examples and Design Patterns

In many practical deployments, LLM agents are embedded in a system, including code writers, data analysts, retrieval modules, safety checkers, and domain-specific advisors. The following examples illustrate how Bayesian structure can be layered on top of LLM-based systems in qualitatively different ways. In each case, LLMs remain non-Bayesian generative models, while a separate Bayesian controller reasons over possibly low-dimensional latent variables tailored to the task.

The first example (Section 4.1) maintains a posterior over a

task outcome for multi-agent code generation. The second example (Section 4.2) infers a hypothesis in a deliberation-style discussion between agents. The third example (Appendix C) learns cross-task competence parameters for routing among agents and tools. Section 4.3 extracts reusable design patterns for Bayesian orchestration from these three examples. Together, these examples and design patterns are not intended as an exhaustive taxonomy or as fully instantiated end-to-end algorithms, but as suggestive starting points for Bayesian multi-agent orchestration in practical deployments and future systems.

### 4.1. Multi-Agent Code Generation And Testing

Consider an AI assistant for software engineering that must generate a Python function from a natural-language specification (Huang et al., 2024; Islam et al., 2025). The assistant consists of a system of $n$ AI agents, indexed by $i \in \{1, \ldots, n\}$. An agent generates candidate code snippets, a retrieval agent provides relevant examples, a static safety checker identifies high-risk code blocks (such as security vulnerabilities or unsafe side effects), and a unit-test runner verifies correctness.

An orchestration framework maintains a belief in a low-dimensional task-level outcome variable $Y$. In this example, $Y \in \{0, 1\}$ represents whether the candidate code passes all tests ($Y = 1$) or fails at least one test ($Y = 0$). Let $i_t \in \{1, \ldots, n\}$ denote the agent queried at call $t$, and let $Z_t$ denote the resulting message. For example, $Z_t$ may be a proposed code snippet, a retrieved reference implementation, or a static-analysis warning. Let $\mathcal{D}_{1:t} = \{(i_s, Z_s) : s \leq t\}$ be the collection of all queried agents and messages observed so far after $t$ agent calls.

Since $Y$ is unknown at decision time, the orchestrator maintains a belief over $Y$. For example, during code development, the orchestrator does not yet know whether the current candidate code would pass the unit tests. The orchestrator's belief is formulated as the posterior mass function $r_t(y) = p(Y = y \mid \mathcal{D}_{1:t})$, $y \in \{0, 1\}$, with prior $r_0(y) = p_0(y)$ before any agent calls. Thus, for each $t$, $r_t(\cdot)$ is a probability distribution over the task outcome $Y$. The belief is defined over the task outcome, not over the parameters or internal mechanisms of any agent. In this way, LLMs are not required to be Bayesian predictors. The Bayesian structure lies in the orchestration layer, which treats each LLM output as defining a likelihood over outcomes rather than over model parameters. Figure 1a provides a schematic view of this orchestration loop and the belief update over the task outcome.

Each agent $i$ can be queried at a known cost $c_i > 0$. From a Bayesian decision-theoretic perspective, the choice of which agent to query next (or when to stop and return code) is an action selected to maximize posterior expected utility under $r_t$, trading off the expected probability of passing

tests against query costs $c_i$ and any safety penalties. An additional agent call is justified only when its expected value of information exceeds its cost. An LLM agent call at time $t$ generates a message $Z_t$ from the queried agent $i_t$.

The relation between $Z_t$ and the outcome $Y$ is described by an observation model $p_{i_t}(z_t \mid y)$ that is learned or approximated from observed outcomes. A new observation updates the orchestrator's belief by Bayes' rule, and the belief update of the outcome given the message is

$$r_t(y) \propto r_{t-1}(y) p_{i_t}(z_t \mid y)^{\alpha_{i_t}}, \tag{1}$$

where $r_{t-1}(y)$ is the belief before the update, and $\alpha_i > 0$ is a reliability weight associated with agent $i$. At time $t$, the update uses $\alpha_{i_t}$, namely the component of the agent-indexed vector $(\alpha_1, \ldots, \alpha_n)$ corresponding to the queried agent $i_t$. The index set is fixed, while the numerical values $\alpha_i$ may be updated online as outcomes are observed. In composite-likelihood terms, each $\alpha_{i_t}$ acts as an exponent on the likelihood factor $p_{i_t}(z_t \mid y)$ and therefore tempers the strength of that agent's likelihood contribution. This factorized form provides an approximate composite likelihood (pseudo-likelihood) update (Varin et al., 2011), and the decomposition into likelihood factors is a modeling choice rather than a unique representation of the true joint likelihood.

The factorization in equation (1) is an approximation that treats agent calls as conditionally independent given the latent outcome $Y$ and the modeled context. If repeated messages are obtained from the same underlying agent, then statistical dependence is expected through shared prompts and internal states. This can be handled by conditioning the observation model on the interaction history or by augmenting the latent state with an agent-specific latent variable that captures shared error structure, so that conditional independence holds given the augmented state.

From a signal processing and control perspective, the orchestrator receiving messages from agents behaves analogously to a Bayesian filter receiving noisy sensor readings (Särkkä, 2014). The sensors themselves need not be Bayesian; it suffices that their observation models $p_{i_t}(z_t \mid y)$ can be estimated from data. The posterior is modeled over the task outcome $Y$. A distribution $q_{i_t}(y \mid z_t)$ is used to approximate the local posterior $p_{i_t}(y \mid z_t)$. In practice, $q_{i_t}(y \mid z_t)$ is obtained by fitting a model on past tasks, using the agent's previous messages and corresponding outcomes. The learned map $z_t \mapsto q_{i_t}(y \mid z_t)$ is not claimed to make the raw message sufficient for $Y$; it defines a decision-oriented approximate belief update. Its adequacy should be judged on held-out tasks by calibration and decision utility, that is, by whether richer message features substantially change task-level predictions or actions. For benchmarks or logs with verifiable outcomes, each task yields pairs $(z_t, y)$ for supervised learning, enabling discriminative fitting and post-hoc

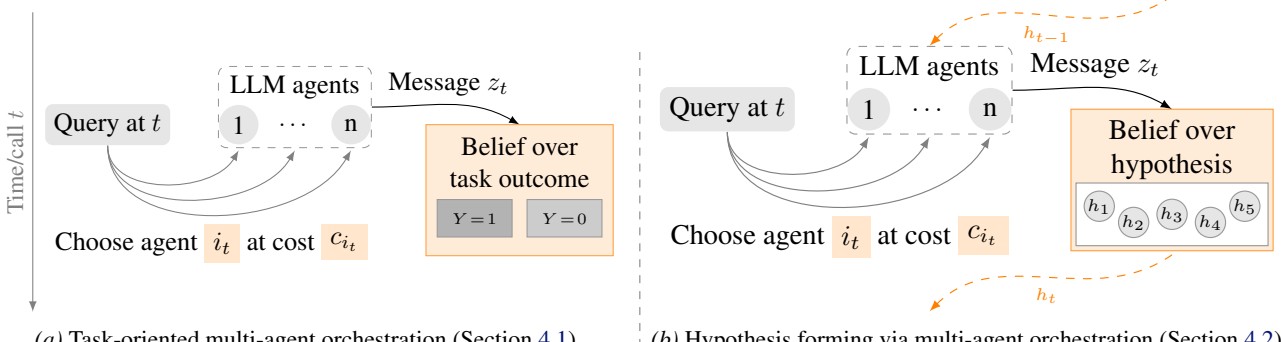

*(a)* Task-oriented multi-agent orchestration (Section 4.1).  *(b)* Hypothesis forming via multi-agent orchestration (Section 4.2).

*Figure 1.* Left: schematic of the example of Section 4.1. At each step $t$, an orchestrator selects an agent $i_t$ at cost $c_{i_t}$, receives a message $Z_t$, and updates a task-level belief state represented by the probability mass function $r_t(y) = p(Y = y \mid \mathcal{D}_{1:t})$ over the binary code-testing outcome $Y \in \{0, 1\}$. Right: schematic of the example of Section 4.2. At each step $t$, an orchestrator selects an agent $i_t$ at cost $c_{i_t}$, receives a message $Z_t$, and updates a belief state over a discrete hypothesis space $\mathcal{H} = \{h_1, \ldots, h_k\}$ for a multi-agent discussion related, for example, to competing scientific explanations or policy options. The belief is represented by $r_t(h) = p(H = h \mid \mathcal{D}_{1:t})$, and it is updated from agent messages treated as noisy observations about which hypothesis is supported.

calibration of $q_i(y \mid z_t)$. In the code-generation setting, this can use pass-versus-fail test outcomes. In other settings, verifier labels, human ratings, or task-completion signals can play the same role. One practical variant is to query an LLM for task-level probabilities or confidence judgments and treat them either as inputs to $q_i(y \mid z_t)$ or as noisy probabilistic observations that are calibrated against observed outcomes. Recent work on agentic uncertainty shows that elicited success probabilities can be systematically overconfident, so such signals should be treated as noisy observations and calibrated before use in control (Kaddour et al., 2026).

Based on the approximate likelihood ratio $\ell_{i_t}(y; z_t) = q_{i_t}(y \mid z_t)/p_0(y)$, the global belief is updated according to

$$r_t(y) = \frac{r_{t-1}(y)\ell_{i_t}(y; z_t)^{\alpha_{i_t}}}{\sum_{y'} r_{t-1}(y')\ell_{i_t}(y'; z_t)^{\alpha_{i_t}}}.$$

If we set the cumulative log-loss of agent $i$ on past tasks as $L_i = \sum_{s:i_s=i} -\log q_i(y_s \mid z_s)$, then the relative reliabilities, seen as an online mechanism for tracking which agents are performing well, can be updated by an exponential-weights scheme, with unnormalized weights $w_i \propto \exp(-\beta L_i)$ for a sensitivity parameter $\beta > 0$ (Freund & Schapire, 1997; Cesa-Bianchi & Lugosi, 2006). These weights can be normalized as $\tilde{w}_i = w_i/\sum_j w_j$ and mapped to likelihood-tempering exponents via a bounded scaling, for example $\alpha_i = \alpha_{\max}\tilde{w}_i$ for some chosen $\alpha_{\max} > 0$. As written, $\alpha_i$ acts only as a tempering exponent in the composite-likelihood update, which can be viewed as a form of generalized Bayes or power-posterior updating (Bissiri et al., 2016). The parameter $\alpha_i$ is distinct from $p_i(z \mid y)$. The observation model captures the nominal evidence in the message, whereas $\alpha_i$ downweights sources that are noisy, miscalibrated, or statistically dependent. In practice, $\alpha_i$ can be tuned on held-out tasks or updated online from outcome

feedback.

This example illustrates how a Bayesian orchestrator can coordinate multiple agents to improve correctness, efficiency, and safety in a single task (code generation). By maintaining a belief over the task outcome and updating it with evidence weighted by agent reliability, the system adapts trust in each agent through reliability weights. The resulting controller can thus be developed to become cost-aware, uncertainty-aware, and able to stop or abstain when confidence is insufficient. This demonstrates how Bayesian reasoning can provide structure to agent interactions even when the underlying agents are not themselves Bayesian. In decision-theoretic terms, the posterior $r_t$ supplies the belief state for expected-utility (or value-of-information) policies that determine which evidence to purchase next and when to stop. For the current example on code generation and testing, the belief state is represented by the posterior distribution $r_t$ over the task outcome $Y$, rather than by a posterior over a hypothesis in a multi-agent deliberation setting.

### 4.2. Multi-Agent Discussion

The deliberation setting in this example differs qualitatively from the example in Section 4.1. The latent variable is now a hypothesis $H$ rather than a code-testing outcome $Y$. Here, $H$ is the task variable that orchestration tracks during deliberation. The hypothesis space may encode logical or causal relations, and agent-specific observation models can support hierarchical priors over agent expertise, or some agents may be more reliable on some subsets of the hypothesis space than on others. The resulting probabilistic model captures both uncertainty over hypotheses and uncertainty over which

agents are informative for which parts of the hypothesis space, instead of focusing solely on task success or failure.

Consider a deliberation setting in which several specialized agents discuss a scientific or policy question on behalf of a user (Du et al., 2024). Each agent is implemented by an LLM with its own system prompt, tools, or training data, for instance, emphasizing different methodological standards or assumptions. The user specifies a question and a hypothesis space $\mathcal{H} = \{h_1, \ldots, h_k\}$ encoding mutually exclusive explanations or policy options.

The interaction proceeds in deliberation steps $t \in \mathbb{N}$. At step $t$, the orchestrator selects an agent $i_t \in \{1, \ldots, n\}$ and receives a message $Z_t$ from $i_t$. Let $\mathcal{D}_{1:t} = \{(i_s, Z_s) \colon s \leq t\}$ be the collection of all queried agents and messages observed after $t$ deliberation steps. A Bayesian orchestrator maintains a belief $r_t(h) = p(H = h \mid \mathcal{D}_{1:t})$ over $h \in \mathcal{H}$ as the discussion progresses, treating the agents' messages as noisy observations about which hypothesis is supported, while the agents themselves are text generators that need not be Bayesian. Figure 1b provides a schematic view of the multi-agent deliberation and the belief update over the hypothesis space.

As a concrete example of multi-agent deliberation, consider incident diagnosis for a production system failure (Luo et al., 2025). The user's question is "what caused the outage", and the hypothesis space consists of a small set of mutually exclusive root causes, for example, $h_1$ hypothesizes a memory leak, $h_2$ a race condition, and $h_3$ an upstream API change. Different agents can be specialized to inspect logs, review recent code changes, or summarize operational context, and their messages $Z_t$ provide noisy evidence about which root cause is supported. The orchestrator updates $r_t(h)$ as the discussion unfolds and decides whether to query another agent, invoke a diagnostic tool, or escalate to a human operator when posterior uncertainty remains high.

The response $Z_t$ of the queried agent $i_t$ admits an observation model $p_{i_t}(z_t \mid h)$, given a hypothesis label $h$ learned from previous deliberation tasks. The orchestrator updates its belief according to

$$r_t(h) \propto r_{t-1}(h) p_{i_t}(z_t \mid h)^{\alpha_{i_t}}, \qquad (2)$$

using agent-specific reliability weights $\alpha_i$ (similarly to the example in Section 4.1), with $\alpha_{i_t}$ denoting the weight of the agent $i_t$ queried at step $t$. The belief update for $r_t(h)$ in equation (2) is analogous to the belief update for $r_t(y)$ in equation (1), with the code-testing outcome $Y$ replaced by the hypothesis $H$.

The belief update of equation (2) is written sequentially in $t$, so the ordering of queries is explicit, and no additional batch factorization assumption is required beyond the chosen observation models. However, if multiple messages are ag-

gregated into a single update by multiplying likelihood factors, then the resulting product should be understood as an approximate composite-likelihood construction, as in Section 4.1.

Decision making in this setting consists of sequentially choosing which agent to query next and when to stop the discussion. A possible rule is to terminate once the posterior places sufficient mass on one hypothesis, for example, when $\max_h r_t(h)$ exceeds a user-specified confidence threshold. More generally, the orchestrator can be framed as solving a Bayesian sequential decision problem; given utilities $u(a, h)$ over final actions $a$ (such as recommending a policy or experimental design) and the true hypothesis $h$, it selects queries to agents in order to maximize expected utility minus information-gathering costs, based on the current posterior $r_t(h)$. If the discussion stops at time $t$, the Bayes action is $a_t^\star = \arg\max_a \sum_{h \in \mathcal{H}} u(a, h) \, r_t(h)$, and further agent queries are warranted only when their expected value of information exceeds their cost. Thus, LLM agents are used only as evidence sources. The uncertainty representation and expected-utility calculations remain in the orchestration layer.

> This example shows how Bayesian reasoning can structure a multi-agent discussion without requiring any individual LLM to be Bayesian. The latent variable is a hypothesis $H$ over explanations or policies, while LLM-based agents provide arguments that are modeled as noisy observations about $H$ with agent-specific reliability. The orchestrator performs posterior updates and sequential query and stopping decisions, in contrast to the previous example of Section 4.1 that focuses on code-testing outcomes. In this way, Bayesian methods drive the interaction and decision layer of a multi-agentic system, treating LLMs as flexible conversational agents whose outputs can be treated as evidence. From a Bayesian decision-theoretic view, $r_t$ is the belief state that specifies the Bayesian action and the value-of-information tradeoff that determines which agent to query next.

### 4.3. Design Patterns For Bayesian Orchestration

Sections 4.1-4.2 and Appendix C can be read not only as examples, but as instances of a common Bayesian template for agentic orchestration. In this template, an orchestrator maintains a belief state over possibly low-dimensional, decision-relevant latent variables, updates that belief from tool and agent outputs through explicit observation models, and selects orchestration actions by posterior expected utility or value-of-information criteria. The design patterns below summarize reusable components of this template, and clarify how Bayesian structure enters at the system level,

even when the underlying LLM components remain non-Bayesian predictors.

1. *Task-level belief factoring.* Represent uncertainty in a latent variable that is directly relevant for orchestration decisions, rather than in token-level uncertainty, for example, a task outcome, a discrete hypothesis, or an agent's competence parameter.

2. *Messages as observations.* Treat each agent or tool output as an observation about the task-level latent variable by specifying an observation model that maps messages to likelihood information used for belief updates.

3. *Reliability-weighted updates.* Introduce source-specific reliability parameters that temper likelihood contributions, so that evidence from weaker or biased sources does not produce overconfident posterior updates, and update reliabilities from outcome-labeled experience.

4. *Dependence-aware evidence pooling.* Account for statistical dependence between evidence sources, for example, repeated prompting of the same model or shared retrieval pipelines, and use conservative pooling or explicit dependence modeling when conditional independence is implausible.

5. *Utility-based and information-based control.* Express routing, stopping, abstention, escalation, and budget allocation as posterior expected-utility decisions or value-of-information decisions, thus making the cost of additional evidence explicit in the orchestration policy.

6. *Cross-task posteriors for routing.* Maintain posteriors over persistent tool or agent profiles across tasks, so that routing can adapt online and trade off exploration against exploitation under uncertainty.

7. *Belief-state distillation.* Maintain a compact belief state as an approximately sufficient statistic of interaction history, so that orchestration remains feasible under context-window and latency constraints while preserving the information needed for subsequent decisions.

## 5. Call To Action

**Benchmarking agentic orchestration.** Outcome-based evaluation is a first step toward principled agentic orchestration, as illustrated by the belief updates over task outcomes and tool competence in Section 4. Benchmarks should be designed so that the orchestrator's latent task variables ad-

mit measurable outcomes, and so that routing, stopping, or escalation to a human expert can be scored jointly with incurred costs. Evaluation should report not only task success but also predictive calibration of task-level beliefs, together with decision-aware criteria that quantify evidence efficiency, for example, via value-of-information metrics. Because realistic orchestration is heterogeneous and often non-stationary, benchmarks should include controlled shifts in task distributions and tool reliability, as well as dependence between evidence channels, so that overconfidence under correlated tool calls is measured rather than obscured by average-case performance.

**Bayesian control-layer modeling.** A second step is to formalize Bayesian control at the level of the agentic orchestrator, by specifying belief states, update rules, and decision rules in forms that can be learned from data with low overhead (to retain feasibility). The examples in Section 4 illustrate design patterns, including a latent state, an observation model that maps tool outputs to likelihood-based information, and a decision rule based on posterior expected utility or value of information. Bayesian agentic AI controllers should be consistent with the properties articulated in Section 2 to remain useful for practitioners. Observation models should be learned and continually recalibrated from interaction logs with measurable outcomes, while tracking uncertainty in the observation model and revalidating under shift. Some components are already accessible with existing tools; task-level priors can often be initialized from historical frequencies or expert defaults, observation models $q_i$ can be learned from outcome-labeled logs, and reliability weights can be updated online. Initialization can come from historical logs or a small warm-start set of labeled tasks, while misspecification under shift can be monitored online with rolling calibration diagnostics or proper scores. When drift is detected, updates should become more conservative via stronger tempering, dependence-aware pooling, abstention, or escalation. More open research challenges include dependence-aware evidence pooling under shared model or retrieval pipelines, robust uncertainty quantification for high-dimensional agent messages under shift, and scalable approximations to multi-step value-of-information in combinatorial orchestration. Because tool calls and repeated prompting produce correlated evidence, update rules should be dependence-aware, for example, via evidence pooling or likelihood tempering, in line with the misspecification and dependence concerns discussed in Appendix A. Finally, orchestrators should standardize decision policies for routing, stopping, escalation, and budget allocation, expressed directly in decision-theoretic terms. To preserve low latency, these policies do not need to use exact value-of-information. Practical implementations can rely on one-step approximations, learned surrogates for expected loss reduction, or amortized controllers trained offline and queried online.

**Engineering and deployment practice.** A third step is to translate Bayesian orchestration into implementations that satisfy the practical properties articulated in Section 2. Agentic orchestrators should expose only simple controls, such as confidence thresholds and cost scales, while keeping belief updates internal and maintaining a manageable context window through distillation of interaction history. Implementations should align with existing typed agent schemas and tool interfaces, so that uncertainty and utilities can be handled with low overhead. Human feedback and multi-agent messages should enter the same update mechanism as probabilistic observations, so that orchestration decisions remain consistent as interactions scale. When human judgments are used, their reliability can be modeled as context-dependent and learned from historical agreement with outcomes. When human and AI opinions are likely correlated, evidence should be pooled conservatively to avoid overconfident updates. Systems should also support logging of beliefs and decisions, enabling inspection of routing and escalation behavior across tasks and modalities.

**Theory for agentic orchestration.** A fourth step is to develop a theory that targets orchestration as a decision problem under uncertainty, rather than prediction only. This includes decision-focused generalization results that quantify how belief-state estimation error and observation-model misspecification propagate into decisions, complementing the decision-theoretic framing in Section 2. Partial observability provides a framework for this setting, aligning with the belief-state updates used throughout Section 4. Robust and generalized Bayes analyses can justify conservative updates, including likelihood tempering and dependence-aware evidence pooling, as principled responses to misspecification, as discussed in Appendix A. Finally, routing across tools fits Bayesian bandit and online-learning frameworks, as in the routing example of Appendix C, yielding regret and sample-complexity guarantees when competence profiles are learned under drift and non-stationarity.

## 6. Conclusion

This position paper argues that Bayesian structure is compelling at the level of *agentic AI control*. LLMs can be treated as powerful predictive components whose outputs serve as evidence for a latent state maintained by a Bayesian controller. This separation aligns uncertainty representation with the quantities that matter for decisions, such as task outcomes, hypotheses, costs, risks, tools, and agent reliability. In this view, the question is not whether an LLM "is Bayesian", but whether the AI system behaves as a coherent decision maker under uncertainty.

Looking forward, Bayesian control offers a unifying framework for several research directions that are important in deployment: validating and improving agent-level belief quality, learning reliability models for tools and agents under distribution shift, designing belief states that compress interaction history while remaining approximately sufficient for subsequent decisions, and building typed interfaces that expose simple user controls. More broadly, the emerging opportunity is to treat agentic AI as *decision-centric*, with probabilistic state, evidence accumulation, and utility-driven policies becoming design primitives. This reframing would optimize the use of Bayesian ideas in the context of decision making in AI systems, allowing LLM research to prioritize predictive capability even if the internal learning and inference dynamics of such models do not always implement Bayes' rule.

A further practical challenge is that the controller's posterior is only as well founded as the observation models and evidence channels used to update it. If LLM and tool outputs are weak or misspecified, or if multiple agent outputs are treated as independent when they are statistically dependent through shared architectures, data, or prompts, then posterior beliefs can become poorly calibrated or overconfident. This understanding highlights the importance of reliability modeling, likelihood tempering, and statistical dependence handling as design choices in Bayesian orchestration of agentic AI systems.

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

# A. Extended Background And Related Work

Bayesian ideas enter agentic machine learning at multiple levels, ranging from implicit belief-state updates learned by meta-training to explicit probabilistic models used for uncertainty quantification and decision making. The current appendix reviews these strands, emphasizing how the practical object of Bayesian modeling shifts from parameter space to function space and, ultimately, to task-level latent variables that are directly relevant for control. The discussion also distinguishes settings in which Bayesian inference is well-posed and empirically testable from the behavior of pretrained LLMs under heterogeneous data and prompting, and motivates Bayesian decision theory as the appropriate interface between uncertainty and action in agentic AI systems.

**Meta-learned policies.** Meta-learning can yield policies whose adaptation behavior approximates Bayesian inference at the level of the agent's internal state, without requiring an explicit posterior over model parameters. Mikulik et al. (2020) show that memory-based meta-learners trained on fast-adaptation tasks behave approximately as Bayes-optimal predictors, in the sense that the meta-trained recurrent state can act as an implicit belief state updated by new evidence. This perspective is further developed by Ortega et al. (2019), who recast memory-based meta-learning as the amortization of sequential Bayesian inference, emphasizing that the learned dynamics can implement a Bayes-filter type of update over task structure. These results support our position by illustrating that Bayesian structure can arise at the level of agentic control and state, even when the underlying components are implemented by deep learning models.

**Posterior over parameters.** One question is whether Bayesian treatments of model parameters remain meaningful in highly overparameterized neural networks. Parameter posteriors are often invoked for model selection, continual learning, and decision making under distribution shift. There is recent evidence that the posterior over parameter space becomes less informative as models grow. Wenzel et al. (2020) demonstrate the "cold posterior" effect, where sampling the parameter posterior degrades performance compared to point estimates. This phenomenon has been scrutinized in follow-up work, with evidence that it can depend on likelihood misspecification and data augmentation choices (Izmailov et al., 2021; Noci et al., 2021; Nabarro et al., 2022). The theoretical explanation may be attributed to geometric symmetries; Sharma et al. (2024); Entezari et al. (2022); Frankle et al. (2020) argue that distinct stochastic gradient decent solutions are often linearly connected (modulo permutation). This implies that apparent modes may be symmetric reflections of a single solution basin, making naive parameter-space diversity deceptive. Altogether, such considerations weaken the case for treating parameter posteriors as the primary object for representing uncertainties in agentic AI systems, and motivate shifting Bayesian structure toward task-level latent variables used for control and decision making.

**Function-space uncertainty and ensembles.** Consequently, attention has shifted from parameter-space posteriors to function-space uncertainty and functional diversity. Wilson & Izmailov (2020) argue that deep ensembles (Lakshminarayanan et al., 2017) succeed by marginalizing over broad solution basins, capturing functional diversity that local Gaussian approximations miss. Even in this setting, feasibility is a practical constraint, since ensembles require multiple trained models and multiple forward passes at inference time, which becomes expensive at LLM scale. However, recent findings suggest that even deep ensembles face limitations at the LLM scale. Abe et al. (2022) show that many of the practical benefits of approximating the posterior through deep ensembles can be realized by a single large model. Similarly, Kirsch (2025) report that functional diversity, and consequently the estimated epistemic uncertainty, collapses for large models due to model-internal ensembling. Additionally, Dern et al. (2025) provide theoretical limitations of ensembles in the overparameterized regime, showing that they may fail to provide reliable uncertainty estimates. These findings reinforce the need to locate uncertainty quantification at the level of agentic states and outputs, rather than relying on internal parameter properties, and motivate latent-state approaches such as world models (Ha & Schmidhuber, 2018; LeCun, 2022). Implicit or latent reasoning methods may change the nature of uncertainty quantification (Hao et al., 2025; Balestriero & LeCun, 2025), but not the need to model agentic AI systems.

**Representing epistemic uncertainty in LLMs.** Separately from the question of parameter posteriors, the representation of epistemic uncertainty for LLM-based systems remains an active and partly unsettled modeling choice. Earlier works such as Malinin & Gales (2021) consider the parameter posterior to represent epistemic uncertainty, yet conduct their experiments on relatively small models by today's standards. More recently, Bakman et al. (2025) define epistemic uncertainty in terms of feature gaps between a model's hidden representations and those of a reference model, rather than as a posterior over parameters. Abbasi-Yadkori et al. (2024) define a metric for epistemic uncertainty based on iterative prompting of the LLM. In particular, a model can be uncertain in the token-level posterior predictive distribution while remaining confident about the underlying semantic answer or task outcome, and conversely, token-level confidence can coexist with uncertainty about the task-level variable that drives decision-making. Together, these observations question whether parameter posteriors, and the

token-level posterior predictive distributions they induce, should be the right primary objects for representing uncertainties in LLM-centric systems.

**In-context learning as Bayesian inference.** At the same time, a number of theoretical works interpret aspects of in-context learning as approximate Bayesian inference in a latent task or concept space. Xie et al. (2022) analyze a setup in which the pretraining corpus consists of text sequences generated by a mixture of hidden Markov models, and they show that achieving low pretraining loss requires the transformer to implement implicit Bayesian updates over a latent concept that underlies each document. In this view, in-context learning corresponds to inferring the latent concept from prompt examples and then predicting under the corresponding posterior. More recent work asks under what conditions LLMs can be made to behave in a more Bayesian way at the level of their outputs (Akyürek et al., 2023). Gupta et al. (2025) show that repeated sampling with Monte Carlo aggregation can yield predictions that approximate Bayes-optimal posterior predictions on structured reasoning tasks, even when the base model's single-shot predictions are biased. Karan & Du (2025) similarly show that inference-time sampling procedures can improve reasoning without additional training, suggesting that approximate Bayesian behavior can emerge at the level of aggregated outputs. These works identify regimes in which Bayesian updates can emerge implicitly in model behavior, but they do not remove the need for an explicit decision-theoretic belief state in agentic system control.

**Transformer's capability of implementing in-context Bayesian inference.** A useful distinction is between controlled regimes in which Bayesian inference is well-posed and testable, and evaluations of pretrained LLMs under heterogeneous data and realistic prompts. In controlled settings, the data-generating process is specified, and the true posterior over the latent task variables is available in closed form, so one can directly test whether a transformer trained on such data can track the Bayes filter. In this regard, Reuter et al. (2025) construct a transformer-based network architecture which, after meta-learning, can emulate Bayesian posterior inference on generalized linear models and latent factor models, with quality close to the state-of-the-art MCMC methods. Very recent work reports that transformers trained on data simulated from a hidden Markov model can match the posterior with high precision (Agarwal et al., 2025a). The authors further analyze the geometric mechanisms by which attention implements the update (Agarwal et al., 2025b).

**Deviations of Bayesian behaviors for pre-trained LLMs.** In contrast, for pretrained LLMs it is often unclear what a single Bayesian target should be at the semantic or task level. Empirical and theoretical critiques indicate, however, that pretrained LLMs and approximate Bayesian deep neural networks may deviate from Bayes' rule when interpreted as full belief-updating agents under generic prompting. Chlon et al. (2025) hypothesize that an LLM's aggregate coding performance across tasks may match information-theoretic Bayesian benchmarks while individual predictions violate classical Bayesian properties because positional encodings break exchangeability. Atwell et al. (2026) introduce Bayesian assessment framework for sycophancy, and use it to show that LLMs often update their stated beliefs in a way that departs systematically from Bayesian benchmarks when user preferences are manipulated. Falck et al. (2024) test LLMs on exchangeable data and show that their in-context predictions violate martingale and exchangeability conditions that are implied by Bayesian posterior predictive processes under standard modeling assumptions. In some cases, such violations can be mitigated by prompting procedures that explicitly encourage exchangeability (Jayasekera et al., 2025). Complementarily, Qiu et al. (2026) show that Bayesian teaching can induce more coherent probabilistic reasoning and updating behavior in LLMs, highlighting that improved belief updating is attainable but not guaranteed by default. These seemingly contrasting results suggest that departures from Bayesian inference can stem from both the heterogeneous pre-training data and the prompting design used for in-context learning, and that Bayesian behavior need not hold uniformly across regimes. For similar reasons, diagnostics of Bayesian behavior, such as exchangeability and martingale constraints, can be sensitive to prompting and model mismatch.

**Deviations of Bayesian behaviors due to approximate inference.** Even when architectures are labeled Bayesian, approximate inference can break Bayes' rule when used for deep neural networks. Pituk et al. (2025) study Bayesian neural networks with common approximate inference schemes and find that they can violate Bayesian sequential updates, for example, by forgetting earlier data or failing to preserve conditional independence. Together with the empirical evidence of non-Bayesian behavior in pre-trained LLMs, these results motivate the present paper's emphasis on placing Bayesian structure in the control layer of an agentic AI system, where the latent variables, observation models, and utilities are defined formally, even when the system's agents (such as pre-trained LLMs or Bayesian neural networks with approximate inference) exhibit Bayesian behavior only in restricted regimes.

**Decision theory and utilities.** Bayesian decision theory couples probabilistic beliefs to actions through utilities and costs, and thus formalizes what it means for an agentic AI system to act rationally under uncertainty (Berger, 1985). In many high-stakes deployments, utilities are structured and context-dependent, and costs need not be linear in money, latency, or risk, so the control layer must represent such tradeoffs rather than relying on token-level likelihood alone. This perspective also

supports explicit value-of-information reasoning, in which additional tool calls or expert queries are warranted only when their expected benefit outweighs their cost. Recent safety-motivated work illustrates this logic by deriving probabilistic constraints based on Bayesian posteriors over hypotheses and cautious (yet plausible) alternatives when screening potentially harmful actions (Bengio et al., 2025). The examples in Section 4 should be read as design patterns for such decision-theoretic control (Amin, 2026), rather than as claims that a single instantiated orchestrator is already a complete deployed solution.

**Scope beyond text modality.** Although the exposition focuses on text-only LLMs, the Bayesian control-layer perspective applies to language models beyond text, including visual language models (VLMs; Alayrac et al., 2022) for images or video, audio language models (Borsos et al., 2023), and multimodal large language models (Chen et al., 2023; Yin et al., 2024) that unify multiple input modalities. Our position therefore concerns Bayesian agentic control and orchestration, and is intended to extend to non-text modalities, treating model outputs as evidence for probabilistic decision making.

## B. Limitations And Mitigation Strategies

**Limitation 1: decision theory under high-dimensional internal representations.** One concern is that Bayesian decision-theoretic control may appear mismatched to agentic AI systems whose internal computations are high-dimensional and opaque, based on LLM-based components, learned world models, or large neural policies. In this view, insisting on an explicit belief state and utility-based control could be seen as an artificial simplification that fails to reflect how effective decision makers operate when their internal representations compress information in complex and task-dependent ways.

**Mitigation for limitation 1.** Bayesian decision theory does not require that the internal mechanism producing candidate actions or evidence be low-dimensional or interpretable. It requires that the system's action selection be coherent with respect to uncertainty and utilities at the level at which decisions are evaluated. Human decision-making is consistent with this separation, since the brain may be viewed as a high-dimensional representation and inference machinery, while the decision is still an action chosen under uncertainty and costs. A similar separation holds in systems such as AlphaGo (Silver et al., 2016) and AlphaZero (Silver et al., 2017; 2018), whose internal state and search are highly complex, yet whose control problem is still a selection among available actions guided by an objective. In this sense, the "lower-dimensional" belief states emphasized in this paper should be read as a typical feasibility-oriented choice, often smaller than raw interaction history, rather than as a requirement imposed by the decision-theoretic framing. Bayesian control is therefore proposed as an interface for action selection, not as a constraint on the dimensionality or structure of internal representations.

**Limitation 2: observation-model specification from agent messages.** A second concern is that Bayesian control appears to rely on specifying observation models that map high-dimensional agent outputs, such as free-form text, retrieved documents, or tool traces, to likelihood information about a task-level latent variable. Since these messages are heterogeneous, context-dependent, and sometimes strategically framed, it may be unclear what the appropriate likelihood should be, how it can be validated, and whether it can remain calibrated under distribution shift. This raises the concern that the Bayesian controller relocates the uncertainty-quantification problem to a difficult modeling step, rather than making it simpler.

**Mitigation for limitation 2.** The Bayesian controller does not need to assume a hand-crafted likelihood. Instead, the observation model can be treated as a statistical component learned from past interactions, where messages are paired with task outcomes or labels that define the latent variable of interest. The model can then be recalibrated as new data arrive, and its influence can be made conservative under misspecification or correlated evidence by introducing reliability weights and likelihood tempering. In this way, the controller uses a learned mapping from messages to task-level evidence, while controlling how strongly any single evidence source can shift the belief state.

**Limitation 3: value-of-information reasoning under tight computational budgets.** A third concern is a potential mismatch between advocating value-of-information control and requiring low overhead. Exact value-of-information reasoning is generally defined through counterfactual comparisons over future observation sequences and decisions, which amounts to solving a nontrivial sequential decision problem. In realistic agentic AI settings with large action spaces and expensive model calls, computing such quantities exactly, or solving the corresponding optimal policy, may be computationally infeasible, so the benefits of value-of-information control could appear to conflict with the feasibility constraints emphasized in the paper.

**Mitigation for limitation 3.** The proposed framework does not require exactly computing the value of information. It uses value-of-information reasoning to formalize the tradeoff between the expected benefit of another query and its cost, and this tradeoff can be implemented with approximations. For example, the controller can compare the cost of an additional tool call to an estimate of the expected reduction in uncertainty under the controller's current belief state, or the expected

reduction in posterior expected loss, using single-step calculations or learned predictors. This retains the intended decision rule while keeping computation light relative to model inference.

**Limitation 4: interpretability and the visibility of uncertainty.** A fourth concern is that introducing a probabilistic belief state may reduce interpretability rather than improve it. If the controller's uncertainty is summarized through a small set of probabilities or a single confidence threshold, then users and developers may lose access to the qualitative reasoning that is often conveyed by the underlying agent's natural-language justification or by the interaction trace. This can create the impression that the system's decision is driven by opaque probabilistic mechanisms, even when the underlying evidence is available in text form.

**Mitigation for limitation 4.** Bayesian control does not require collapsing uncertainty to a single scalar, and it need not replace interaction traces or natural-language explanations. The belief state is used for action selection, while the system can still retain and present the underlying evidence, together with a decomposition of the uncertainties and costs driving the decision. This can increase clarity by separating evidential support from the decision rule, rather than hiding decisions inside text.

## C. Bayesian Routing In Agentic Systems

In the two examples of code generation (Section 4.1) and discussion among agents (Section 4.2), the central latent variable is task-specific, namely a binary code-testing outcome $Y$ or a hypothesis $H$ in a multi-agent deliberation. Agent messages are modeled as observations about the current task, and Bayesian inference is local to that task. In the present routing example, the primary latent structure concerns cross-task properties of agents and tools (Fu et al., 2022). Each new task contributes evidence about which agents are likely to succeed, and the resulting posterior over agent profiles is carried across tasks to balance exploration and exploitation in future routing decisions. The uncertainty is therefore about agents themselves, rather than about a single task outcome.

For each task, an agentic AI system must decide which agent or tool to invoke, possibly in sequence, with varying costs, latencies, and areas of expertise. A Bayesian routing controller treats this as a sequential decision problem in which past performance informs which agents are likely to succeed on new tasks. The underlying LLMs and tools remain black-box output generators. The Bayesian structure lives in a controller that maintains uncertainty over agent profiles and uses that uncertainty to guide routing.

Tasks arrive at times $t \in \mathbb{N}$. Each task is described by observable features $x_t$ (for example, a representation of the user query, domain, or difficulty), and the router chooses an agent $i_t \in \{1, \ldots, n\}$ to handle it. After execution, the system observes an outcome $y_t$ measuring task quality, such as binary success or a scalar reward. Here, $y_t$ denotes the task outcome, as in Section 4.1. The difference is in the timing of observation. In Section 4.1, the outcome $Y$ is unknown during the sequence of agent calls and is therefore treated as latent until it is eventually revealed by running unit tests. In the current routing setting of this appendix, $y_t$ is observed after the chosen agent or tool executes on the task at time $t$, so each task provides an observed label $y_t$.

Each agent $i$ has latent parameters $\psi_i$ in the routing model that determine its success probability as a function of task features, through a likelihood model $p(y_t \mid x_t, i_t = i, \psi_i)$. The parameters $\psi_{1:n} = (\psi_1, \ldots, \psi_n)$ play the role of cross-task latent variables.

The controller maintains a joint posterior

$$p(\psi_{1:n} \mid x_{1:t}, \mathcal{D}_{1:t}) \propto p(\psi_{1:n}) \prod_{s=1}^{t} p(y_s \mid x_s, i_s, \psi_{i_s}), \tag{3}$$

where $x_{1:t} = \{x_s \colon s \leq t\}$ is the feature sequence, and $\mathcal{D}_{1:t} = \{(i_s, y_s) \colon s \leq t\}$ is the routing history. For a new task with features $x_{t+1}$, the posterior induces a predictive success probability

$$\pi_i(x_{t+1}) = \mathbb{E}[p(y_{t+1} = 1 \mid x_{t+1}, i, \psi_i) \mid x_{1:t}, \mathcal{D}_{1:t}]$$

for each agent $i$, which the router can combine with a cost $c_i$ and user-specified utilities to choose $i_{t+1}$ via a Bayesian bandit or model-selection approach.

In this example, a Bayesian bandit formulation is a natural instantiation because the router maintains a posterior over competence parameters $\psi_{1:n}$, and sampling policies convert this posterior into an exploration-exploitation rule in a decision-theoretic way. For example, Thompson sampling selects actions by sampling $\psi_{1:n}$ from the posterior and acting optimally

for the sampled parameters, which is Bayes-optimal under the assumed prior in classical settings and admits regret analyses under broad conditions (Thompson, 1933; Gittins et al., 2011; Russo & Van Roy, 2014). Bayesian bandits provide a principled framework for using uncertainty over $\psi_{1:n}$ to achieve cost-aware routing and adaptation across tasks.

As a concrete example, consider a support assistant that routes recurring user requests across a small tool ecosystem, for example a code agent, a retrieval agent, and a data-analysis agent, with different costs and response times. Each incoming request has observable features $x_t$ (such as code snippets, a data-analysis specification, or response times in the interaction), and the system observes an outcome $y_t$ after execution, for example whether the user accepted the solution or whether a code check passed. The router's uncertainty is about which agents tend to succeed on which types of requests, and this uncertainty is refined across tasks as outcomes accumulate, yielding a posterior over agent competence that supports cost-aware routing, with routing decisions improving over time rather than relying on a fixed heuristic.

The likelihood factorization of equation (3) is a modeling assumption, often reasonable when tasks are treated as conditionally independent draws given the feature sequence $x_{1:t}$, the chosen agents $i_{1:t}$, and the agent profiles $\psi_{1:n}$. More structured models can capture temporal dependence or non-stationarity if needed.

Decision rules in this setting take the form of Bayesian bandit policies. Given utilities $u(y, i)$ and costs $c_i$, the controller can select $i_t$ to approximately maximize the posterior expected utility $\mathbb{E}[u(y_t, i_t) - c_{i_t} \mid x_{1:t}, \mathcal{D}_{1:(t-1)}]$, with terms for exploration, such as Thompson sampling (Thompson, 1933) based on $p(\psi_{1:n} \mid \mathcal{D}_{1:(t-1)})$. Because the model operates in a possibly low-dimensional space of agent-level parameters and task features, it remains compatible with short-interaction workflows. Each routing decision is a single-step computation based on current posteriors, and all learning occurs incrementally as outcomes are observed. Throughout, LLM-based agents and tools are treated as system components that provide outputs and observed success signals, without requiring these components to be Bayesian. The Bayesian framework determines how routing preferences are updated and how agent heterogeneity is exploited.

While the formulation of equation (3) focuses on single-step routing decisions $i_t$, many agentic tasks require the sequential composition of multiple tools. In such combinatorial action spaces, exact Bayesian search or multi-step planning becomes computationally expensive. To maintain the low-latency requirements of modern agents, the controller can use *amortized Bayesian inference*. For example, generative flow networks (GFlowNets; Bengio et al., 2023) or amortized variational methods can be trained to sample high-utility action trajectories from an approximate target distribution over trajectories, such as a utility-tempered posterior, providing a scalable path from simple routing to complex sequential orchestration.

> This example illustrates Bayesian routing as a form of across-task learning over agents and tools, rather than within-task inference over a single latent outcome. The controller maintains a posterior over agent-level (competence) parameters $\psi_{1:n}$ that are updated from observed successes and failures across tasks, and uses this posterior to solve a Bayesian exploration-exploitation problem when selecting which agent to call next. In contrast to earlier examples that infer a code-testing outcome or hypothesis for a task, the Bayesian routing model learns structured priors and posteriors over the tool ecosystem itself. LLMs and tools can thus be non-Bayesian predictors, while the orchestration layer uses Bayesian reasoning to allocate tasks efficiently, manage costs and latencies, and adapt routing decisions as evidence about agent behaviour accumulates.

