# OpenReview forum: "Position: Agentic AI Orchestration Should Be Bayes-Consistent"
_ICML.cc/2026/Position_Paper_Track — ICML 2026 Position Paper Track regular_

### Official Review · Reviewer_2fCA · 2026-02-23

**Significance:** 4
**Argument Clarity:** 4
**Rating:** 5
**Confidence:** 4

**Questions:**

1. The framework relies on observation models that map agent messages to likelihoods over task outcomes. How are these models calibrated initially, and how does the system detect and adapt when the observation model becomes misspecified due to distribution shift?
2. Exact value-of-information computation requires solving a nested sequential decision problem (computing the expected utility of the optimal policy after a hypothetical observation). Given the paper's emphasis on "negligible latency," what specific approximations or amortized strategies do you advocate for estimating VoI in real-time orchestration?
3. The paper states that human feedback should enter "as probabilistic observations within the same Bayesian structure." However, human-provided labels or preferences are typically treated as ground truth or rankings in RLHF, not as noisy observations with quantified uncertainty. How do you propose to elicit or estimate the reliability (likelihood function) of human feedback in real-time, particularly when human experts may have varying competence across different hypothesis spaces or when feedback is sparse and delayed? Do you envision a separate meta-learning process for human reliability parameters, and how do you prevent the system from becoming overconfident when human and AI agent opinions correlate?

**Alternative Views Section:**

Yes

**Compliance With Llm Reviewing Policy A Conservative:**

Affirmed.

**Discussion Potential:**

3

**Final Justification:**

My concerns have been adequately addressed, with very good answers and paper modifications.

**Paper Summary:**

This position paper argues that agentic AI systems should implement Bayesian decision-making at the control layer while treating underlying LLMs as black-box predictive components. The authors distinguish between token-level prediction (handled by LLMs) and task-level decision-making under uncertainty (handled by a Bayesian controller), asserting that these require different uncertainty representations. The proposed framework entails: (i) A Bayesian orchestrator maintains explicit belief states over low-dimensional, decision-relevant latent variables. (ii) These belief states are updated via observation models calibrated against measurable outcomes. (iii) Actions are selected by maximizing posterior expected utility or comparing the value of information against costs; examples of actions include querying specific agents, routing tasks, or stopping. The authors illustrate this framework with three examples: multi-agent code generation (maintaining beliefs over binary test outcomes), multi-agent deliberation (maintaining beliefs over discrete hypotheses), and Bayesian routing (maintaining beliefs over cross-task agent competence). From these, they extract reusable design patterns including task-level belief factoring, reliability-weighted updates, and dependence-aware evidence pooling.

**Position:**

Yes

**Position In Title:**

Yes

**Related Work:**

4

**Strengths And Weaknesses:**

Strengths

The paper presents a rigorous conceptual framework that identifies a genuine gap in current practice: the mismatch between syntactic uncertainty (next-token probabilities) and semantic/task-level uncertainty required for decision-making. The separation of concerns—between prediction (LLMs) and decision-making (Bayesian controller)—is logically consistent and addresses the "cold posterior" and epistemic uncertainty collapse issues noted in recent literature. The paper is well-structured, the bibliography is extensive and current, and it effectively addresses credible opposing positions.

Weaknesses

While the paper emphasizes "negligible latency," the proposed framework requires maintaining posterior distributions, updating reliability weights via exponential weighting schemes, and potentially hierarchical Bayesian modeling over observation parameters. In latency-sensitive production environments where agent calls already take hundreds of milliseconds, the overhead of Bayesian updates (even if lightweight relative to LLM inference) may still be non-negligible compared to simple greedy heuristics.

**Support:**

3

---

> ### Author Rebuttal · Authors · 2026-03-27
>
> Thank you for the careful and supportive review. We appreciate your clear summary of the paper's position and your questions. To address them, we made several edits to make the mitigations clearer in the main text.
>
> First, we agreed with your point that "negligible latency" was too strong as written. A more accurate statement is that the controller adds low overhead relative to model or tool inference rather than negligible cost. To address this, in Section 2, in the list of desirable properties, we revised that sentence to:
>
> "It should do so with low additional latency and low memory overhead relative to model or tool inference, and it should yield fewer redundant tool calls and fewer incorrect or unsafe actions at a given target risk level".
>
> This better reflects our claim, not that exact Bayesian control is free, but that practical control-layer updates should be lightweight relative to the model/tool calls they regulate, and justified by improved routing, stopping, and escalation decisions.
>
> Second, regarding initial calibration and misspecification under shift, observation models can be warm-started from measurable outcomes, monitored online, and made conservative when drift is detected. To clarify this, in Section 5 ("Bayesian control-layer modeling"), immediately after the sentence on continual recalibration, we added:
>
> "In practice, initialization can come from historical logs or a small warm-start set of labeled tasks, while misspecification under shift can be monitored online with rolling calibration diagnostics or proper scores. When drift is detected, updates should become more conservative via stronger tempering, dependence-aware pooling, abstention, or escalation".
>
> Regarding your first question, the observation model is warm-started from measurable outcomes, monitored online, and made conservative when shift is detected.
>
> Third, regarding real-time value of information, the paper advocates approximate or amortized decision rules rather than exact nested sequential planning. To clarify this, in the same subsection we added:
>
> "To preserve low latency, these policies do not need to use exact value-of-information. Practical implementations can rely on one-step approximations, learned surrogates for expected loss reduction, or amortized controllers trained offline and queried online".
>
> This is the approximation regime we had in mind. Our position does not require solving a full nested sequential decision problem.
>
> Fourth, regarding human feedback, human judgments can be treated as context-dependent noisy evidence with learnable reliability and conservative pooling under correlation. To clarify this, in Section 5 ("Engineering and deployment practice"), immediately after the sentence on treating human feedback as probabilistic observations, we added:
>
> "When human judgments are used, their reliability can be modeled as context-dependent and learned from historical agreement with outcomes. When human and AI opinions are likely correlated, evidence should be pooled conservatively to avoid overconfident updates".
>
> So yes, we envision separate source-reliability parameters for humans, with conservative pooling when human and AI evidence are correlated.
>
> We are grateful for these questions. They helped us clarify that the paper's practical regime is approximate, low-overhead, and conservative under misspecification, rather than exact Bayesian planning at every orchestration step.

---

> > ### Author Rebuttal · Reviewer_2fCA · 2026-04-01
> >
> > very good answers and paper modifications

---

### Official Review · Reviewer_qiab · 2026-03-09

**Significance:** 3
**Argument Clarity:** 2
**Rating:** 4
**Confidence:** 3

**Questions:**

Q1: Could the authors clarify under what conditions the sufficiency gap between q_i(y | z_t) and the full message z_t is small enough that the resulting posterior r_t(y) remains well-calibrated?
Q2: Could the authors clarify which components of the proposed architecture — observation model construction, prior specification, reliability weight estimation, and value-of-information approximation — they regard as straightforward engineering problems solvable with existing tools, and which they regard as genuinely open research challenges requiring new methodology?

**Alternative Views Section:**

No

**Compliance With Llm Reviewing Policy A Conservative:**

Affirmed.

**Discussion Potential:**

3

**Final Justification:**

I have confirmed that the authors' rebuttal has solved my concerns and is persuasive. Thus, I have raised my score by 1 point.

**Paper Summary:**

This position paper argues that the orchestration and control layer of agentic AI systems should make decisions that are at least consistent with Bayesian decision theory. The paper proposes a separation between prediction and decision: LLMs serve as black-box evidence sources whose outputs are treated as probabilistic observations, while a dedicated Bayesian controller maintains a belief state over low-dimensional, task-relevant latent variables, updates it via reliability-weighted composite likelihood, and selects actions by posterior expected utility or value-of-information criteria. The position is compelling insofar as it identifies a genuine and underexplored mismatch between token-level predictive uncertainty and the system-level epistemic uncertainty that drives agentic decisions, and proposes a principled framework for bridging this gap without modifying the underlying LLM components.

**Position:**

Yes

**Position In Title:**

Yes

**Related Work:**

3

**Strengths And Weaknesses:**

One of the paper's most valuable contributions is its rigorous articulation of the distinction between token-level predictive uncertainty and system-level epistemic uncertainty. An LLM's token distribution can be sharply peaked while the system remains genuinely uncertain about the task-level latent variable that drives action selection, and conversely, a diffuse token distribution can coexist with high confidence about the underlying task outcome — a misalignment that provides a principled justification for locating Bayesian structure in the orchestration layer rather than inside the LLM itself. The Bayesian framework is furthermore well-suited to the structure of the problem, defining the orchestrator as a controller that maintains a belief state over low-dimensional decision-relevant latent variables, updates it via reliability-weighted composite likelihood, and selects actions by posterior expected utility or value-of-information criteria.

The paper's central weakness lies in the coarseness of the observation model on which its Bayesian machinery depends. Because the true observation model over the high-dimensional message space is intractable, the framework substitutes a discriminative approximation. This substitution is statistically valid only when the substitute approximation constitutes a sufficient statistic for y given z_t, a condition that is almost certainly violated in practice: a code snippet or a deliberation argument contains rich structural information about y and the resulting sufficiency gap causes the posterior r_t(y) to be a miscalibrated projection of the true posterior rather than the true posterior itself. The framework provides no principled argument for why Bayesian coherence with respect to this coarse and misspecified summary should yield better decisions than non-Bayesian alternatives.

A notable limitation, however, is that the paper stops short of actionable guidance. It articulates what a Bayesian agentic system should do but does not specify how to construct the observation model, elicit priors, set reliability weights, or approximate value-of-information quantities in practice. The design patterns remain at a level of abstraction that leaves the critical engineering decisions unresolved, and the paper's central claims remain hypotheses deferred to future empirical work.

**Support:**

3

---

> ### Author Rebuttal · Authors · 2026-03-27
>
> Thank you for the careful and constructive review, and especially for highlighting the distinction between token-level and task-level uncertainty; that distinction is central to our position. We found your two questions particularly helpful, and we made revisions to clarify both the approximation induced by $q_i(y\mid z_t)$ and the intended scope of the paper.
>
> First, regarding the sufficiency-gap concern, the controller uses a decision-oriented approximate belief update whose adequacy can be judged empirically by calibration and downstream decision quality. To clarify this, in Section 4.1 we added:
>
> "The learned map $z_t\mapsto q_{i_t}(y\mid z_t)$ is not claimed to make the raw message sufficient for $Y$; it defines a decision-oriented approximate belief update. Its adequacy should be judged on held-out tasks by calibration and decision utility, that is, by whether richer message features substantially change task-level predictions or actions".
>
> Answer to Q1: the sufficiency gap is acceptable only to the extent that richer representations of $z_t$ do not substantially improve task-level calibration, abstention behavior, or decision utility on deployment-matched validation tasks. Otherwise, the controller should be treated as misspecified and the update made more conservative, for example via stronger tempering or abstention.
>
> Second, regarding the paper's level of actionability, we would like to distinguish which ingredients are near-term engineering and which remain open research challenges. To clarify this, in Section 5 ("Bayesian control-layer modeling") we added:
>
> "Some components are already accessible with existing tools; task-level priors can often be initialized from historical frequencies or expert defaults, observation models $q_i$ can be learned from outcome-labeled logs, and reliability weights can be updated online. More open research challenges include dependence-aware evidence pooling under shared model or retrieval pipelines, robust uncertainty quantification for high-dimensional agent messages under shift, and scalable approximations to multi-step value-of-information in combinatorial orchestration".
>
> Answer to Q2: prior initialization, supervised calibration of $q_i$, and online reliability tracking are the parts we view as near-term engineering, whereas dependence-aware pooling, shift-robust uncertainty, and scalable multi-step value-of-information remain open research problems.
>
> Third, regarding scope, the examples are design patterns rather than fully instantiated end-to-end algorithms; accordingly, at the start of Section 4 we revised the sentence to:
>
> "Together, these examples and design patterns are not intended as an exhaustive taxonomy or as fully instantiated end-to-end algorithms, but as suggestive starting points for Bayesian multi-agent orchestration in practical deployments and future systems".
>
> More generally, we agree that a full end-to-end construction with guarantees would be valuable. Our claim in the paper is more modest; when the uncertainty that matters for action lives at the task, agent, or tool level, making that uncertainty and the utility tradeoffs explicit in the controller is a principled design target, even if the resulting belief state is approximate. We are therefore not claiming universal dominance over all non-Bayesian alternatives under arbitrary misspecification. Rather, the point is that Bayesian control makes uncertainty and dependence auditable. *Since this is a position paper, we aimed to articulate that target and provide plausible design patterns rather than a complete main-track algorithm. We believe the added text makes both the scope and the level of concreteness clearer.*

---

> > ### Author Rebuttal · Reviewer_qiab · 2026-04-03
> >
> > I would like to thank the authors for their detailed rebuttal. Most of my initial concerns have been addressed; however, the fundamental motivation for why a Bayesian framework is strictly necessary in this context remains somewhat opaque.
> >
> > I agree with Reviewer cjyG that the current positioning of the paper still feels slightly like a "strawman" argument. While I believe there are inherent advantages to utilizing Bayesian methodologies at the orchestration level—consistent with their utility in broader statistical applications—these strengths are not yet clearly articulated in the manuscript.
> >
> > I strongly encourage the authors to explicitly highlight these specific advantages in the final version to better justify their methodological choice and strengthen the paper’s overall contribution.

---

### Official Review · Reviewer_cjyG · 2026-03-10

**Significance:** 3
**Argument Clarity:** 4
**Rating:** 5
**Confidence:** 4

**Questions:**

Do you see a connection to works that apply probability elicitation techniques for LLMs, in an attempt to quantify the internal beliefs or predictions of LLMs in the same way expert knowledge is quantified in decision-support systems? Is this something the orchestrator could be doing as well, querying LLMs specifically to gain better calibrated uncertainty estimates etc, instead of constructing black-box estimators for the likelihood as in Section 4.1?

**Alternative Views Section:**

Yes

**Compliance With Llm Reviewing Policy A Conservative:**

Affirmed.

**Discussion Potential:**

3

**Final Justification:**

I had no major issues with the paper before the rebuttal, and the authors addressed my minor concerns well, for instance changing the paper title to better match the exact position they are arguing for (and fixed the formatting). The paper makes overall a clear position on a timely and important topic, with no obvious weaknesses.

**Paper Summary:**

The authors argue that agentic AI systems making autonomous decisions should make the following standard decision theoretic formulation, levering the established literature on Bayes-optimal decisions. Moreover, they argue that this should be done at the level of the AI system orchestrating multiple LLMs and tools, instead of attempting to make LLMs themselves Bayesian. The authors sketch how this could be done and provide concrete examples that illustrate the concept in varying scenarios.

**Position:**

Yes

**Position In Title:**

Yes

**Related Work:**

4

**Strengths And Weaknesses:**

The paper takes a clear position and provides good argumentation on why we would want AI systems to be consistent with Bayesian decision theory. However, this is somewhat of a lazy premise -- it is hard to imagine why anyone would actually be against this as a general goal. There is much more freedom on *how* that is achieved and the actual position of the paper is more related to this, but this is not seen in the title. I marked that criterion as "no" because I believe something along the lines of "Bayes-consistent decisions for agentic AI systems should be done at the orchestration level" would better communicate the actual position in a stand-alone manner.

I think my point is largely confirmed by the Alternative Views section, with half of the content being about alternative technical solutions for achieving this, either by trying to make Bayesian LLMs or by changing slightly the decision-theory formulation. The only real alternative that is not about doing Bayes-constent decisions, heuristics and prompting, is a bit of a strawman -- people are doing that because they can do that, not because they believe it would be better. Nevertheless, the discussion about Bayesian LLMs is certainly valid and I can imagine people wanting to challenge what is proposed here by supporting the view that we can make the LLMs Bayesian as such. Some people could certainly argue we should at eventually get there.

The presentation is very clear, actively helping the reader to follow the main arguments and positioning the paper well with the related literature. The proposed elements are clearly and sharply communicated but remain on the level suitable for a position paper, listing aspects that should be taken into account but not yet telling exactly how. Section 4 is well framed as example design patterns, giving idea of generality despite and outlining scenarios of different nature. However, this part is somewhat detached from the ML literature that makes following the paper feel slightly non-scientific in places (e.g. "this factorized form provides an approximate composite likelihood update" would call for a reference discussing how this kind of things can be done in practice). The call-to-action and conclusions are clear, and the additional literature discussed in Appendix is extremely broad and covers the topic well, so the issue of missing references in Section 4 is just a presentational thing. The appendix adds to the value of the paper as a discussion-started and general reference for people interested in the topic -- there are already many concrete ways someone could advance this position by working on any of these aspects.

**Support:**

3

---

> ### Author Rebuttal · Authors · 2026-03-27
>
> Thank you for the thoughtful and positive review. We appreciate your title suggestion, your framing of the paper's actual position, and your question about probability elicitation.
>
> First, regarding the paper's stand-alone positioning, we agreed that the title should state more explicitly that the Bayes-consistent decision making we advocate occurs at the orchestration level; to this end, we changed the title to: "Position: agentic AI orchestration should be Bayes-consistent". We believe this better communicates the actual stand-alone position of the paper.
>
> Second, regarding the presentation of heuristic and prompting approaches, they are primarily practical engineering baselines rather than an opposed stance. To clarify this, in Section 3 under "Heuristic or prompting-based approaches" we added: "We view these approaches primarily as practical engineering baselines and current defaults, rather than as an opposed stance".
>
> Third, regarding probability elicitation, we view elicited probabilities as an evidence channel for the orchestrator rather than an alternative to orchestration-level Bayesian control. To clarify this, in Section 4.1, immediately after the sentence introducing $q_i(y\mid z_t)$, we added: "One practical variant is to query an LLM for task-level probabilities or confidence judgments and treat them either as inputs to $q_i(y\mid z_t)$ or as noisy probabilistic observations that are calibrated against observed outcomes".
>
> This is very much in line with our position. Our view is that elicited probabilities are not an alternative to the orchestration-level controller, but one possible way of constructing its evidence channel. The key point is that even elicited probabilities still need calibration, dependence handling, and coupling to utilities/costs at the control layer.
>
> Finally, regarding the presentation of Section 4, to anchor the composite-likelihood update more explicitly in the statistical literature, in Section 4.1 we revised the relevant sentence to: "This factorized form provides an approximate composite-likelihood (pseudo-likelihood) update~\citep{varin2011overview}, and the decomposition into likelihood factors is a modeling choice rather than a unique representation of the true joint likelihood".
>
> We are grateful for these suggestions. They helped us sharpen both the positioning and the presentation of the paper.
>
> ---
>
> Reference mentioned in our response:
>
> @article{varin2011overview,
>   author = {Varin, Cristiano and Reid, Nancy and Firth, David},
>   title = {An overview of composite likelihood methods},
>   journal = {Statistica Sinica},
>   volume = {21},
>   number = {1},
>   pages = {5--42},
>   year = {2011}
> }

---

> > ### Author Rebuttal · Reviewer_cjyG · 2026-04-02
> >
> > I acknowledge having read the rebuttal, which addresses my minor concerns well. I think the new title conveys the position better, and the additional clarifications regarding elicitation and composite likelihoods are sound. I have no open concerns regarding the paper.

---

### Official Review · Reviewer_4BtR · 2026-03-16

**Significance:** 4
**Argument Clarity:** 3
**Rating:** 5
**Confidence:** 4

**Questions:**

See above;

Line 252-260, why $\alpha_i$ is in exponent? Can this be learnt? why $p_{i_t} (z_t |y)$ doesn’t model it already?

Can you bring a brief discussion of limitation and mitigation in the main paper?

**Alternative Views Section:**

Yes

**Compliance With Llm Reviewing Policy A Conservative:**

Affirmed.

**Discussion Potential:**

3

**Paper Summary:**

The paper argues that in agentic AI systems, Bayesian decision-making should be applied at the control and orchestration layer, rather than making the parameters of LLMs themselves Bayesian. By the control layer, the authors mean quantifying uncertainty over decision variables in the task/decision space and using it to guide how LLMs and tools are used, e.g., when to query them, when to stop, how to route tasks, and how to allocate budgets. The paper suggests separating prediction (what LLMs do well) from decision-making under uncertainty (what the orchestration layer should handle). The motivation is that LLMs do not reliably satisfy Bayesian updating properties (Falck et al., 2024), and that parameter posteriors become less informative at large scale (Kirsch, 2025). The authors propose seven desirable properties for Bayesian control layers and illustrate the approach with three examples: (1) multi-agent code generation with a posterior belief over task outcomes Y, (2) multi-agent deliberation with a posterior over hypotheses, and (3) Bayesian routing. The paper concludes with a call for benchmarks, formalized control-layer modeling, engineering practices, and Bayesian theory for agentic AI.

**Position:**

Yes

**Position In Title:**

Yes

**Related Work:**

3

**Strengths And Weaknesses:**

Strengths:
- The paper is well-written and clearly structured; I enjoyed reading it.
- The paper was thoughtful throughout while presenting desired specifications, alternative viewpoints, illustrative examples, and limitations.
- The three examples demonstrating how agentic AI systems can leverage Bayesian decision-making is beneficial.

Weakness:
- For examples, how could one estimate the observation model or liklihood if you use any existing coding or other LLM benchmark?
- If one models LLM as truly Bayesian network (ignore computational challenge for now), in this case there will also be a latent uncertainity on the task/decision space right? Why that uncertainity is not worse as compared to modelling in decision space as presented in this position?

**Support:**

3

---

> ### Author Rebuttal · Authors · 2026-03-27
>
> Thank you for the positive assessment and for the constructive questions. We appreciate your comments on the examples and on the need to make the practical side clearer. To address your points, we made four edits.
>
> First, regarding how the observation model can be estimated in practice, it can be learned from outcome-labeled interactions on tasks with measurable task-level outcomes. To clarify this, in Section 4.1, immediately after the sentence "In practice, $q_{i_t}(y\mid z_t)$ is obtained by fitting a model on past tasks...", we added:
>
> "For benchmarks or logs with verifiable outcomes, each task yields pairs $(z_t,y)$ for supervised learning, enabling discriminative fitting and post-hoc calibration of $q_i(y\mid z_t)$. In the code-generation setting, this can use pass-versus-fail test outcomes. In other settings, verifier labels, human ratings, or task-completion signals can play the same role."
>
> Regarding your first question, the observation model is learned from outcome-labeled interactions, using benchmark or deployment tasks for which the task-level outcome is measurable.
>
> Second, regarding the role of the exponent $\alpha_i$, it is a separate robustness and reliability control rather than something already absorbed by the nominal observation model. To clarify this, in Section 4.1, immediately after the sentence introducing generalized Bayes, we added:
>
> "The parameter $\alpha_i$ is distinct from $p_i(z \mid y)$. The observation model captures the nominal evidence in the message, whereas $\alpha_i$ downweights sources that are noisy, miscalibrated, or statistically dependent. In practice, $\alpha_i$ can be tuned on held-out tasks or updated online from outcome feedback."
>
> So yes, $\alpha_i$ can be learned. Our intent is that $p_i(z \mid y)$ models the message-to-outcome relation, while $\alpha_i$ is a separate robustness parameter for source reliability, misspecification, and dependence that one may not want to force entirely into the observation model.
>
> Third, regarding the relation between a hypothetically Bayesian LLM and our position, control has to be defined in decision space because orchestration is evaluated at the level of task outcomes, costs, and actions. To clarify this, in Section 3 under "Bayesian LLM compatibility with our position", immediately after the sentence "The uncertainties may arise at the task level and can be utility-dependent", we added:
>
> "Accordingly, even if a fully Bayesian LLM induced uncertainty over task variables, decision-space modeling would remain a useful control object because it is aligned with the latent variables, costs, and actions on which orchestration is evaluated."
>
> We want to make the distinction that even in that hypothetical setting, Bayesian uncertainty inside the LLM would not by itself replace a control-layer belief state, because the orchestration problem is defined at the level of task outcomes, utilities, and actions.
>
> Finally, regarding limitations in the main paper, it is helpful to surface the central practical limitation and its mitigation in condensed form without reproducing the appendix; accordingly, in Section 2 we replaced the appendix-only pointer with:
>
> "A key limitation is that observation models derived from high-dimensional agent messages can be misspecified, and repeated tool calls can yield correlated evidence. The practical mitigation is to make updates conservative via recalibration from measurable outcomes, likelihood tempering, dependence-aware evidence pooling, and abstention or escalation when posterior confidence is fragile. Appendix B elaborates on these and other limitations and mitigations."
>
> We are grateful for these suggestions. They helped us make various aspects of the paper more concrete.

---

> > ### Author Rebuttal · Reviewer_4BtR · 2026-04-03
> >
> > I asked a followup question but waiting for authors response. Since ack has a deadline, i am ack it now.
> >
> >
> >
> > <EDITED: Reposting the same comment from 3 days ago; perhaps there is a bug in open review that restricts asking  questions from authors>
> >
> > thank you for the response.
> >
> > >  $\alpha_i$ downweights sources that are noisy, miscalibrated, or statistically dependent.
> >
> > I guess you mean that it's more of an aleatoric noise. Since $\alpha_i$ are unknown, how would you know that it affect exponentially? Isn't a more principled way of modelling it is $p_i(z \mid y) p(\alpha | ....)$ or something like this?

---

### Decision · Program_Chairs · 2026-04-30

**Decision:**

Accept (regular)

**Comment:**

Overall, this paper is a strong, well-structured position paper that identifies a genuine conceptual gap in agentic AI decision-making and is highly recommended as a valuable discussion-starter for the community. The reviewers broadly praised the logical separation of prediction from decision-making and found the illustrative design patterns beneficial for demonstrating multi-agent code generation, deliberation, and routing. However, the authors must address several critical weaknesses highlighted across the reviews, most notably the lack of actionable guidance on constructing and calibrating the coarse observation models, the computational overhead and latency concerns of Bayesian updates in production environments, and the need to justify why Bayesian coherence on a misspecified summary yields better decisions. Despite these implementation-level limitations, the conceptual framework and extensive referencing make it a compelling and significant contribution.